# Ulnar Nerve Management in Complex Elbow Dislocations: A Retrospective Monocentric Study

**DOI:** 10.3390/jpm14111076

**Published:** 2024-10-26

**Authors:** Carlotta Faccenda, Elisa Dutto, Francesco Bosco, Alessandro Dario Lavia, Bruno Battiston

**Affiliations:** 1Hand Surgery and Reconstructive Microsurgery Department, CTO Hospital, A.O.U. Città della Salute e della Scienza, 10126 Turin, Italy; 2Hand Surgery Department, Ospedale San Paolo, Via Genova 30, 17100 Savona, Italy; 3Department of Precision Medicine in Medical, Surgical and Critical Care (Me.Pre.C.C.), University of Palermo, 90133 Palermo, Italy; 4Department of Orthopaedics and Traumatology, G.F. Ingrassia Hospital Unit, ASP 6, 90131 Palermo, Italy; 5Department of Economics, Boston College, Boston, MA 02467, USA

**Keywords:** ulnar nerve, ulnar nerve transposition, complex elbow dislocations

## Abstract

Background/Objectives: The ulnar nerve’s unique anatomy makes it vulnerable to complex elbow dislocations. Depending on the nature of the injury, the clinical treatment and outcomes related to the nerve may vary. Unfortunately, the current literature provides limited and fragmented information on managing the ulnar nerve and the incidence of neuropathy in complex elbow dislocations. This study aimed to determine the occurrence of ulnar nerve pain and its relationship with transposition. Methods: A retrospective evaluation was conducted on a consecutive series of 44 patients who underwent surgery for complex elbow dislocations. The average follow-up period was 29 months. Patients were categorized based on their condition (trans-olecranon fracture–dislocation, Terrible Triad, Monteggia-like lesions, and injuries not falling into the previous categories). The study assessed whether the ulnar nerve was released from the cubital tunnel and underwent transposition. Additionally, the study examined the number of patients experiencing ulnar pain in the postoperative period and its duration over time. All patients were also evaluated using the Mayo Elbow Performance Score (MEPS), Disabilities of Arm, Shoulder, and Hand (DASH) score, and Oxford Elbow Score (OES). Results: Patients who underwent simultaneous ulnar nerve transposition surgery with complex elbow dislocation showed a higher incidence of neuropathy. In these patients, the symptoms were less severe but lasted longer. MEPS, DASH, and OES did not show statistically significant differences between the two groups; however, they were slightly better for the group without ulnar nerve transposition. Conclusions: Surgeons should refrain from routinely transposing the ulnar nerve in complex elbow dislocations. However, further studies involving larger populations are necessary.

## 1. Introduction

Elbow fracture–dislocations present a formidable challenge in orthopedic surgery due to their intricate nature, involving the management of both bone fractures and associated soft tissue injuries [1,2,3]. Such injuries demand a high level of surgical expertise and careful decision-making to balance the dual goals of restoring stability and maintaining flexibility in the elbow joint. These injuries are further complicated by the frequent occurrence of fractures alongside dislocations, which appear in approximately one-quarter of cases, adding layers of complexity to both diagnosis and treatment [1,4].

These complex dislocations often impact crucial medial ligament structures, most notably the coronoid process, which is implicated in about 60% of these cases. The damage to these structures typically necessitates a meticulous medial surgical approach. This approach must be strategically planned to address the vulnerable anatomical components without compromising the ulnar nerve, which is particularly at risk during such procedures [5,6].

The medial collateral ligament (MCL) of the elbow is composed of three distinct parts: the anterior bundle MCL (AB-MCL), the posterior bundle MCL (PB-MCL), and the transverse ligament. Each plays a pivotal role in elbow stability. The AB-MCL, arising from the medial epicondyle, extends to the sublime tubercle, providing critical stabilization against valgus stress. The PB-MCL, also originating from the medial epicondyle, attaches to the olecranon, forming the structural basis of the cubital tunnel, thereby positioning the ulnar nerve in close proximity to potential injury sites, especially in the event of trauma or surgical intervention [6,7] (Figure 1).

The operative strategy typically involves a medial approach, designed to enable a thorough visualization and meticulous repair of the injured ligamentous structures. Such procedures often address avulsions of the ligament complex from the epitrochlea with humeral reinsertion using robust fixation methods such as suture anchors or trans-osseous sutures. Central to the surgery is the careful handling of the ulnar nerve, including its identification, neurolysis, and possibly a cubital tunnel release to mitigate the risk of postoperative neuropathic complications. In some instances, an ulnar nerve transposition may be necessary to optimize patient outcomes and reduce the risk of nerve entrapment post-surgery [7,8].

Despite the frequency and severity of these injuries, the literature detailing the incidence of ulnar nerve neuropathy in the context of complex elbow dislocations remains surprisingly limited. This deficiency extends to the guidance available on managing the ulnar nerve during such surgeries, which this study aims to address. Our investigation will thoroughly analyze the incidence of ulnar nerve neuropathy associated with these injuries and evaluate the role of the subcutaneous anterior transposition of the ulnar nerve in managing such cases.

We hypothesize that there will be no significant difference in the incidence of ulnar nerve neuropathy between treatment modalities involving the subcutaneous anterior transposition of the ulnar nerve and those that do not employ this technique. Our study is designed to rigorously assess the effectiveness of this surgical modification, with the goal of providing definitive, evidence-based recommendations for the management of the ulnar nerve in patients suffering from complex elbow dislocations. Through comprehensive data collection and analysis, we aim to furnish orthopedic surgeons with crucial insights into the optimal strategies for reducing ulnar nerve complications, thereby enhancing patient outcomes and contributing to the broader field of orthopedic trauma care.

## 2. Materials and Methods

### 2.1. Study Design

This study adopted a retrospective monocentric approach, meticulously analyzing a cohort of 44 consecutive patients treated for complex elbow dislocations at the Orthopedic Trauma Centre of Turin between 2017 and 2020. The cohort was managed by the Hand Surgery and Reconstructive Microsurgery team, underscoring a comprehensive examination of clinical outcomes.

Specifically, the research focused on comparing two distinct patient groups: those who underwent ulnar nerve subcutaneous anterior transposition and those who did not, during the surgical management of complex elbow dislocations. The primary objective was to assess the incidence of neuropathy linked directly to the surgical procedures employed. This involved a detailed comparison of postoperative nerve afflictions in both groups to ascertain whether ulnar nerve transposition constitutes a significant risk factor for the development of neuropathic complications.

A rigorous evaluation protocol was established to monitor the onset and progression of ulnar sensory and motor neuropathies post surgery. The severity of these neuropathies was meticulously classified based on McGowan’s classification system [9], which categorizes neuropathies into three distinct grades: Grade 1, characterized by subjective sensory disturbances; Grade 2, which includes muscle weakness potentially accompanied by atrophy of the intrinsic muscles; and Grade 3, marked by severe sensory–motor impairments culminating in paralysis of the ulnar musculature.

Further analysis extended to the examination of recovery rates from neuropathy and the frequency of reoperations necessary among the studied cohorts. In addition to these clinical evaluations, the study incorporated a comparative analysis using well-established functional outcome measures such as the Mayo Elbow Performance Score (MEPS), Disabilities of Arm, Shoulder, and Hand (DASH) score, and Oxford Elbow Score (OES) [10,11,12]. These tools provided a quantitative framework to assess functional recovery and long-term disability following surgical treatment, offering a holistic view of patient outcomes post-intervention.

### 2.2. Inclusion and Exclusion Criteria

The inclusion criteria targeted patients older than 16 years who underwent surgical treatment. Complex dislocations were intricately categorized based on specific anatomical disruptions. The terrible triad, a critical classification, involved a combination of elbow dislocation, radial head fracture, and a coronoid process fracture. Another category was the trans-olecranon fracture–dislocation, which included dislocations accompanied by a fracture through the olecranon. Additionally, the Monteggia-like lesion was characterized similarly to the classical Monteggia fracture–dislocation but involved a proximal ulnar shaft fracture coupled with a radial head dislocation. To accommodate the diverse presentations of complex elbow dislocations, an additional category was established for cases that did not fit the previous categories. This category included dislocations where there was a fracture of the radial head, coronoid, epitrochlea, epicondyle, or distal humerus. The exclusion criteria for the study were patients with preoperative ulnar neuropathy, those whose surgical approach did not involve the ulnar nerve, and those with incomplete historical documentation. All participants included in the study had consented to be part of the research.

### 2.3. Surgical Technique

The surgical approaches to the elbow in this study encompassed posterior, medial, and bilateral methods. These techniques were meticulously chosen based on the specific requirements of each case, aiming to provide optimal visibility and accessibility while minimizing tissue disruption.

A critical component of these procedures involved the careful handling of the ulnar nerve within the cubital tunnel. The identification and subsequent management of this nerve are paramount, given its significant implications for elbow functionality. Initially, the ulnar nerve was delicately released from several constrictive structures: the cubital tunnel itself, Osborne’s arcuate ligament, and Struther’s arch, as shown in references [6,7]. This release was performed to ensure that no undue pressure or entrapment would compromise nerve integrity.

Throughout the surgery, the nerve was safeguarded using a vascular loop. This technique allowed for the nerve to be mobilized without applying traction or direct handling, thereby reducing the risk of nerve damage. This aspect of the procedure is crucial for maintaining the nerve’s viability and function post-operation.

At the conclusion of the fracture–luxation-related interventions in the cohort that did not undergo nerve transposition, the ulnar nerve was meticulously repositioned back to its anatomical location in the cubital tunnel. This repositioning is vital to maintain the natural biomechanics of the elbow and ensure nerve health.

Conversely, in cases requiring nerve transposition, a novel and robust method was employed. A fascial flap was carefully crafted from the flexor–pronator muscle group. This flap, anchored at the medial epicondyle, was designed to create a new pathway for the ulnar nerve. The nerve was then transposed anteriorly to lie within this fascial envelope, which was subsequently sutured to the subcutaneous tissue layers. This newly created “neo-pulley” system is designed to facilitate the free gliding of the nerve, which is crucial for preventing post-surgical complications such as nerve entrapment or tension.

For those patients in whom the nerve was transposed, the original cubital tunnel was closed using resorbable sutures, thus securing the area and promoting healing. The decision as to whether to transpose the nerve or not was made based on the surgeon’s clinical judgment, taking into consideration the individual’s specific anatomical and pathological conditions to optimize surgical outcomes and recovery trajectories.

This comprehensive approach to managing the ulnar nerve during elbow surgeries underscores the complexity and precision required in such procedures. It highlights the importance of tailored surgical strategies that adapt to the nuances of individual patient anatomy and clinical presentations, ensuring both the functionality and longevity of the surgical results.

### 2.4. Data Extraction

The research was conducted utilizing our institution’s advanced electronic medical database as the primary source of data. This comprehensive system enabled the systematic collection of both demographic and procedural characteristics using a meticulously designed template. The collected data encompassed a wide range of variables, including the duration of follow-up, patient gender, age at the time of surgery, and specific details regarding the side of the body that was affected. Furthermore, it detailed the nature of the fractures—whether they were open or closed—as well as the presence of injuries to the ipsilateral upper extremity, the incidence of polytrauma, and the precise type of injury sustained. Treatment modalities were recorded along with the critical decision of whether the ulnar nerve required transposition. Additional insights were gleaned from follow-up visits and physical examinations, focusing on postoperative distress of the ulnar nerve, recovery trajectories, and the necessity of any reinterventions.

To accurately classify the injuries, preoperative radiographs and CT scans housed in the electronic database were meticulously analyzed. This phase of data classification was spearheaded by two of the study’s authors (CF, ED), who not only oversaw the data collection process but also engaged directly in the evaluation and treatment of patients. These evaluations included administering the MEPS, DASH, and OES, which are critical metrics in assessing functional outcomes. In instances of analytical discrepancies or interpretational differences, a third author (BB) was consulted to reach a consensus, ensuring the integrity and accuracy of the interpretation and classification of data. This rigorous methodology underscores our commitment to delivering a scientifically robust analysis that contributes meaningful insights into the treatment outcomes for patients suffering from complex elbow injuries.

### 2.5. Data Analysis

The statistical analysis was performed using STATA software, version 17 (2021). A t-test was conducted to compare two patient data groups: one was treated with subcutaneous transposition of the ulnar nerve and the other without ulnar nerve transposition. The aim was to test whether these two groups had statistically significant differences in neuropathy onset and scores. The scores used for comparison included MEPS (MEPS pain, MEPS mobility, MEPS function), DASH, and OES. The analysis involved combining the data by group (no ulnar surgery and ulnar surgery) and utilizing *t*-test analysis. The null hypothesis assumed that the difference in means between the two groups was not statistically significant. It is important to note that the t-test assumed that the variances among the two populations were the same. A *p*-value below 0.05 was used as the threshold to reject the null hypothesis, which stated that the difference in means was not statistically significant.

## 3. Results

### 3.1. Study Cohort and Demographic Characteristics

From the initial cohort, 57 patients underwent surgery for complex elbow dislocation. Exclusions were as follows: one patient due to preoperative ulnar nerve involvement, seven patients whose surgical approach did not affect the ulnar nerve, and five patients lost to follow-up (including one death unrelated to the trauma). This left 44 patients for the final analysis.

Among these, the injuries were categorized as follows:Twenty-four terrible triadsThree trans-olecranon fractures-dislocationsFour Monteggia-like injuriesThirteen cases uncategorized due to the presence of multiple concurrent fractures. These included: six coronoid fractures, one olecranon fracture, one epicondyle fracture, one fracture of the coronoid and lateral humeral condyle, three radial head fractures, and one humeral epicondyle fracture.

Out of these forty-four patients, eight underwent subcutaneous anterior transposition of the ulnar nerve during surgery, while the remaining thirty-six did not. Figure 2 provides a flowchart summarizing patient enrollment and the occurrence of neuropathy post surgery.

The mean age of the patients was 48 (minimum 16, maximum 77) years. The mean follow-up was 29 months (minimum 13, maximum 48) months, and most patients were male (65.9%). Table 1 presents the main demographic characteristics of the included patients, which are distinct for the two groups.

### 3.2. Clinical Outcomes Evaluation

Postoperatively, ten patients showed ulnar nerve disorders, and three of these patients received subcutaneous transposition. Neuropathy was also classified according to McGowan’s classification. Spontaneous regression, the need for a second intervention, and persistent patterns were also evaluated. The data are summarized in Table 2.

This analysis encompasses 44 cases, comprising 36 patients without ulnar nerve transposition surgery and 8 patients who underwent ulnar nerve transposition during surgery.

Upon investigating the distinction between patients who underwent ulnar surgery and those who did not, a non-significant difference (*p*-value > 0.05) emerges, particularly in reported ulnar discomfort among the surgery group. Despite the non-significant level, there was a visible prevalence of patients reporting suffering among the group who underwent surgery.

An evaluation of scores, including MEPS, DASH, and OES, indicates no statistically significant differences in means between the two groups (no ulnar surgery and ulnar surgery). The *p*-values for these measures exceed 0.05, suggesting that the observed differences might be coincidental. Notably, patients without prior ulnar surgery exhibited more favorable outcomes.

The anterior subcutaneous transposition of the ulnar nerve has a potential negative impact on the assessed measures, including MEPS, DASH, and OES. These findings contribute modestly but perceptibly to our comprehension of the potential consequences of ulnar nerve surgery on patient outcomes, with possible implications for treatment decisions and patient management strategies. The data are summarized in Figure 3.

## 4. Discussion

The main findings of our study are a significantly lower incidence of ulnar nerve complications and more favorable outcomes in MEPS, DASH, and OES among patients with complex elbow dislocations who did not undergo ulnar nerve transposition during surgical intervention. Specifically, among the cohort of 44 consecutive patients treated for complex elbow dislocations, the incidence of ulnar nerve neuropathy was 22.7%. This rate of neuropathy is consistent with previous findings reported in the literature regarding ulnar nerve sufferance associated with distal humerus fractures, reinforcing our observations’ relevance within the broader context of orthopedic research [8,13,14,15,16,17].

Our study further stratified the patients into subgroups based on the type of ulnar nerve management employed during surgery. In the subgroup where patients underwent subcutaneous transposition of the ulnar nerve, the incidence of nerve sufferance was notably higher, at 37.5%. Conversely, the subset of patients without specific ulnar nerve surgery exhibited a significantly lower incidence rate of 19.5%. This stark contrast underscores the potential impact of surgical techniques on patient outcomes, suggesting that the avoidance of ulnar nerve transposition might be beneficial in managing complex elbow dislocations.

Among the patients with nerve suffering, two with nerve transposition and three without transposition resulted in spontaneous resolution, while three patients of the second group had a secondary transposition. There were two persistent cases of neuropathy, one in each group; both patients had refused further surgery.

Nerve suffering can occur preoperatively, intraoperatively, or postoperatively. Damage to the nerve may be direct damage caused by the fracture or related to handling, traction, contact with hardware, transposition, and scar tissue formation. In our study, suffering was attributable to intraoperative and postoperative causes, as we excluded patients with preoperative pathology.

In the recent literature, there are contrasting opinions regarding treating ulnar nerve suffering with elective surgery; some authors claim there is no difference between simple nerve release and transposition [18,19,20], while others support the idea that simple release is effective and has fewer risks [21,22,23]. Most authors find more significant benefits in performing nerve transposition in cases of arthrolysis due to elbow stiffness, probably related to the tension generated on the nerve due to the increased range of motion (ROM) [24,25,26].

Clain et al. report increased ulnar nerve suffering in patients undergoing nerve transposition during ulnar collateral ligament reconstruction compared to non-transposed patients [27].

The current literature indicates that ulnar neuropathy occurs in approximately 20% of patients suffering from distal humerus fractures [28]. The evidence concerning systematic nerve transposition is nuanced; while it typically shows no therapeutic benefit, it can sometimes pose a risk factor for the development of neuropathy. Notably, transposition has been shown to be beneficial in cases where preoperative neuropathy is already present, suggesting a targeted approach in surgical decision-making [8,29,30,31].

Despite the relatively high incidence of neuropathy in humerus fractures, the literature remains deficient in reporting the overall prevalence of neuropathy in complex elbow dislocations. This gap underscores a significant need for further research. Current guidelines on managing the ulnar nerve in these contexts are sparse and exhibit a high degree of variability and contradiction across different studies and reports [29,30,31].

In addressing terrible triad injuries, Toros et al. [32] recommend performing prophylactic nerve release to prevent the development of ulnar neuropathy. Contrasting with this view, Gupta et al. [33] argue against the routine use of prophylactic ulnar nerve decompression, highlighting the lack of consensus among practitioners.

For trans-olecranon injuries involving significant structural challenges, Luen-go-Alonso et al. [34] suggest using a posterior approach without explicitly identifying the ulnar nerve, simplifying the surgical process. However, this technique’s safety is debated, as evidenced by Shin et al. [5], who report multiple instances of ulnar nerve injuries following such procedures in traumatic elbow situations.

Elbow fracture dislocations represent a complex category of injuries that require intricate and often immediate decision-making from the attending surgeons. Based on our comprehensive analysis of the existing studies [5,8,29,30,31], we conclude that while statistical significance is lacking, the preponderance of evidence leans towards avoiding immediate ulnar nerve transposition during the primary surgical intervention. Such an approach minimizes the risk of further nerve damage and aligns with a more conservative and patient-specific surgical strategy.

## 5. Future Research Directions

As we continue to explore the complexities of ulnar nerve management in elbow dislocations, it is crucial to expand the breadth and depth of our investigations to refine and optimize treatment strategies [35,36]. One critical area for future research is the inclusion of a more diverse and extensive patient cohort. Such an expansion would enhance the generalizability of our findings and enable a more detailed understanding of how different demographics respond to various surgical interventions.

Longitudinal studies with extended follow-up periods are equally essential. These studies will allow us to evaluate the long-term outcomes and sustainability of surgical treatments, providing insights into the efficacy and durability of nerve management techniques over time. Additionally, there is a significant need for randomized controlled trials to compare the effectiveness of different surgical approaches, such as nerve transposition versus nerve release. These comparative studies are vital to establishing robust, evidence-based guidelines that can systematically guide clinical practice [36,37].

The evolution of surgical tools and techniques also presents a promising frontier. The development of minimally invasive procedures, coupled with the integration of advanced imaging technologies, offers the potential to reduce operative trauma and enhance recovery outcomes. Such technological advancements could revolutionarily improve how we manage the ulnar nerve during complex elbow surgeries [36].

Moreover, fostering multi-center collaborations could accelerate the accumulation of data, enabling more comprehensive analyses and potentially more definitive conclusions about the optimal management strategies. These collaborative efforts can bridge the gaps in the current knowledge and unify treatment protocols across different regions and institutions.

It is also imperative to shift some of our focus towards patient-centered outcomes. Future research should prioritize the clinical efficacy of treatments and their impact on patient quality of life, including pain management, functional recovery, and overall satisfaction. Developing new metrics or utilizing existing patient-reported outcome measures could provide a more holistic view of the benefits and limitations of our surgical interventions.

Lastly, understanding the economic implications of various treatment options through cost-effectiveness analyses is crucial. Such studies would offer valuable insights for healthcare providers and policymakers, assisting in making informed decisions about resource allocation and treatment accessibility.

## 6. Limitations

Our study presents several limitations that should be acknowledged. Firstly, the retrospective design inherently limits our ability to establish causality. Secondly, as the research was conducted at a single center, this confines the number of participants and the generalizability of our findings to broader populations. Thirdly, the surgical approach, specifically the choice of transposition, was occasionally influenced by the preferences of individual surgeons, which could introduce bias in the selection of treatment modalities. Lastly, the small sample size restricted our ability to perform subgroup analyses based on the specific types of complex elbow dislocation, which might have provided more detailed insights into the effectiveness of different treatment strategies. Additionally, the limited diversity and small size of the patient cohort may have impacted the results’ statistical power and generalizability. Future studies should aim to include a more extensive and diverse patient population to improve the robustness and applicability of the findings across broader demographics. Furthermore, conducting a randomized controlled trial would be essential for directly comparing the outcomes of different surgical approaches, thereby providing more substantial evidence to guide optimal treatment strategies.

## 7. Conclusions

This study assessed the efficacy and outcomes of ulnar nerve transposition in the treatment of complex elbow dislocations. Our findings revealed no statistically significant differences in the mean outcome scores between the patient groups with and without ulnar nerve transposition. This lack of difference, however, was coupled with a trend towards more favorable outcomes in patients who did not undergo the procedure, suggesting that the benefits of avoiding ulnar nerve transposition may outweigh the advantages of performing it. The consistent pattern of better outcomes in the non-transposition group underscores the potential for improved patient recovery without the additional risks associated with surgical manipulation of the ulnar nerve. Given these findings, we advise against the routine use of ulnar nerve transposition in managing elbow fracture–dislocations. The decision to transpose the ulnar nerve should be reserved for specific cases where individual patient factors and surgical judgment deem it necessary. Overall, our study supports a more conservative approach to ulnar nerve management in the context of elbow dislocations.

## Figures and Tables

**Figure 1 jpm-14-01076-f001:**
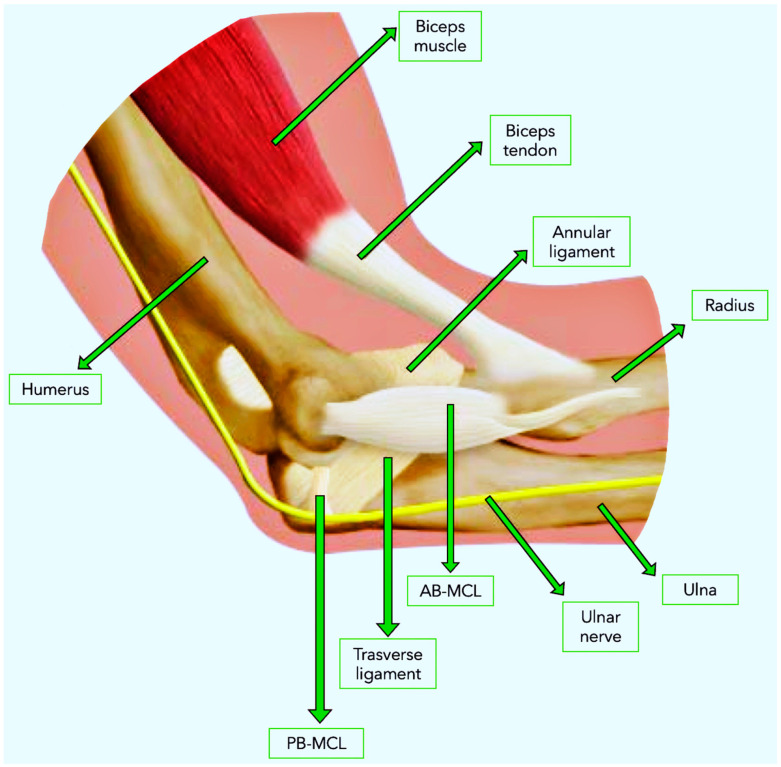
Anatomical representation of the elbow, highlighting the main ligaments (annular, AB-MCL, PB-MCL, and transverse), the biceps tendon, and the ulnar nerve. The bone structure is shown, consisting of the humerus, radius, and ulna. AB-MCL: anterior bundle–medial collateral ligament, PB-MCL: posterior bundle–medial collateral ligament.

**Figure 2 jpm-14-01076-f002:**
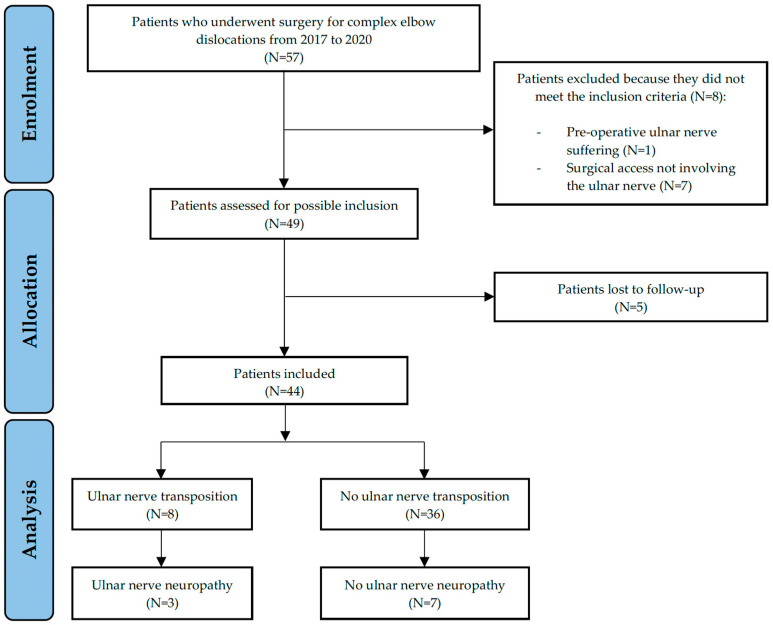
Flowchart of patient enrollment in the study and the onset of neuropathy.

**Figure 3 jpm-14-01076-f003:**
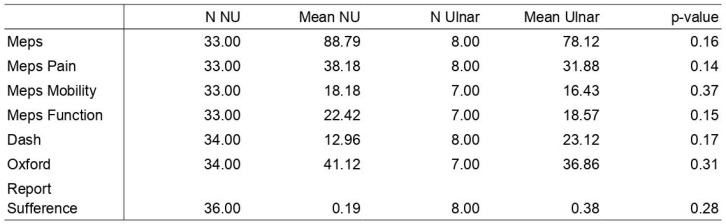
Primary outcomes of patient’s evaluation. N NU = number of patients not undergoing transposition; Mean NU = average values of non-transposition patients; N Ulnar = number of patients undergoing transposition; Mean Ulnar = average values of transposition patients.

**Table 1 jpm-14-01076-t001:** Demographic characteristics of patients, which are distinct for the two groups.

Variables	Transposition	No Transposition
**Age, years (Mean ± SD)**	48 ± 15.2	49 ± 18.4
**Sex**		
Males, N (%)	5 (62.5%)	24 (66.7%)
Females, N (%)	3 (37.5%)	12 (33.3%)
**Fracture-dislocation types**		
Terrible triad, N (%)	7 (87.5%)	17 (47.2%)
Trans-olecranon, N (%)	0	3 (8.3%)
Monteggia-like	0	4
Others	1 (12.5%)	12
**Follow-up, months (Mean ± SD)**	32.1	28.3

% = percentage, N = number of evaluation cases; SD = standard deviation.

**Table 2 jpm-14-01076-t002:** Trends in ulnar nerve suffering in patients who underwent or did not undergo subcutaneous nerve transposition.

	Transposition (N = 8)	No Transposition (N = 36)
**Ulnar neuropathy ***		
Grade 1, N	1	4
Grade 2, N	2	2
Grade 3, N	0	1
**cNeuropathy regression**		
Spontaneous, N	2	3
Secondary transposition, N	0	3
Persistent, N	1	1

N = number of evaluation cases; * = Ulnar neuropathy classified according to McGowan.

## Data Availability

Data are stored in the internal arthroplasty registry of Hand Surgery and Reconstructive Microsurgery Department, CTO Hospital, A.O.U. Città della Salute e della Scienza, 10126 Turin, Italy.

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
