# Peer review of "Ulnar Nerve Management in Complex Elbow Dislocations: A Retrospective Monocentric Study"

_jpm, 2024, doi:10.3390/jpm14111076_

Round 1
Reviewer 1 Report
Comments and Suggestions for Authors
The paper investigates the occurrence of ulnar nerve pain and looks at whether ulnar nerve transposition should be routinely performed in such cases.
The topic is highly relevant to the field of orthopedic surgery, especially in the context of managing elbow fracture-dislocations, which are complex injuries. The paper fills a specific gap in the lit. by providing more structured insights into the outcomes of ulnar nerve management during complex elbow dislocations. The research contributes to a nuanced area where limited and fragmented data is currently available, especially regarding the incidence of neuropathy following these procedures
It adds to the subject area by focusing specifically on the role of ulnar nerve transposition during complex elbow surgeries, presenting data from a monocentric retrospective cohort. The findings suggest that transposition doesnt significantly improve patient outcomes and may anctually increase the risk of postop neuropathy. These insights offer good guidance to orthopedic surgeons, challenging previous recommendations that advocate for routine nerve transposition
The conclusions are consistent with the evidence and arguments presented. the conclusion is supported by the comparative analysis, which found no statistically significant differences in outcomes between the two patient groups.
I’d suggest the below improvements:
- The introduction would benefit from including more recent references to strengthen the context.
- a larger, more diverse patient cohort to increase the statistical power and generalizability of results.
- Include a randomized controlled trial in future studies to directly compare the outcomes of different surgical approaches.
- The tables and figures effectively summarize the data however, adding more detailed charts or visuals of the statistical analysis, particularly regarding the comparison between the 2 groups, would improve clarity. A kaplan Meier survival curve could also be added to demonstrate the timeline of neuropathy progression or recovery across the cohort.
- The results section could be clearer, particularly with improved visual aids and more concise presentation of the outcome data.
- Further controls could include stratifying patients by additional variables eg preop health status or severity of injury.
- While the conclusions are supported by the findings, emphasizing the study's limitations, especially the small sample size, would enhance the discussion.
- The references are appropriate and cover a range of relevant studies in the field however, authors should consider adding more recent studies or meta-analyses to reinforce the discussion on nerve transposition outcomes
The manuscript is generally understandable but would benefit from moderate editing for clarity, sentence structure, and grammatical precision. Some sentences are overly complex, and the flow of certain sections can be improved for better readability. There are also occasional awkward phrases that could be streamlined to enhance the overall quality of the language.
Author Response
|
Numbered Reviewer Remark and Manuscript Line Number |
Author Response |
Revised Manuscript Line Number and Text Change |
|
The paper investigates the occurrence of ulnar nerve pain and looks at whether ulnar nerve transposition should be routinely performed in such cases. The topic is highly relevant to the field of orthopedic surgery, especially in the context of managing elbow fracture-dislocations, which are complex injuries. The paper fills a specific gap in the lit. by providing more structured insights into the outcomes of ulnar nerve management during complex elbow dislocations. The research contributes to a nuanced area where limited and fragmented data is currently available, especially regarding the incidence of neuropathy following these procedures. It adds to the subject area by focusing specifically on the role of ulnar nerve transposition during complex elbow surgeries, presenting data from a monocentric retrospective cohort. The findings suggest that transposition doesnt significantly improve patient outcomes and may anctually increase the risk of postop neuropathy. These insights offer good guidance to orthopedic surgeons, challenging previous recommendations that advocate for routine nerve transposition. The conclusions are consistent with the evidence and arguments presented. the conclusion is supported by the comparative analysis, which found no statistically significant differences in outcomes between the two patient groups. |
Thank you for positive feedback on our research |
/ |
|
I’d suggest the below improvements: |
We have added recent references as suggested. |
Please see the added references:
- Baltassat, A., Baldairon, F., Berthe, S., Bellier, A., Bahlouli, N., & Clavert, P. Creation of a replicable anatomic model of terrible triad of the elbow. J Orthop Surg Res. 2024;19(1):638. - Soderlund, T., Zipperstein, J., Athwal, G. S., & Hoekzema, N. Monteggia Fracture Dislocation. J Orthop Trauma. 2024;38(9S):S26-S30. - Giai Via, R., Faccenda, C., Artiaco, S., Dutto, E., Lavia, A. D., Massè, A., & Battiston, B. Functional and subjective outcomes after surgical management of complex elbow dislocations: a retrospective study. Eur J Orthop Surg Traumatol. Published online September 20, 2024. doi:10.1007/s00590-024-04103-5. |
|
- a larger, more diverse patient cohort to increase the statistical power and generalizability of results. |
This point has been included among the study’s limitations and provides us with the opportunity to continue data collection in order to update the findings already highlighted in this study. |
Please see the edited text:
“Lastly, the small sample size restricted our ability to perform subgroup analyses based on the specific types of complex elbow dislocation, which might have provided more detailed insights into the effectiveness of different treatment strategies. Additionally, the limited diversity and small size of the patient cohort may have impacted the results' statistical power and generalizability. Future studies should aim to include a more extensive and diverse patient population to improve the robustness and applicability of the findings across broader demographics.“ |
|
- Include a randomized controlled trial in future studies to directly compare the outcomes of different surgical approaches. |
As suggested, this concept has been added to the final part of the manuscript. |
Please see the edited text:
“Furthermore, conducting a randomized controlled trial would be essential for directly comparing the outcomes of different surgical approaches, thereby providing more substantial evidence to guide optimal treatment strategies.” |
|
- The tables and figures effectively summarize the data however, adding more detailed charts or visuals of the statistical analysis, particularly regarding the comparison between the 2 groups, would improve clarity. A kaplan Meier survival curve could also be added to demonstrate the timeline of neuropathy progression or recovery across the cohort. |
Thank you for your valuable comments. The tables and figures currently included in the manuscript already present the obtained data in detail, aiming to summarize and clarify the main results of the study. However, regarding the Kaplan-Meier curve, it was not included because the small number of patients in our sample could negatively affect the statistical evaluation and reliability of such a curve. We believe that with such a limited sample size, it would not be possible to obtain a reliable representation of neuropathy progression or recovery. We remain available for any further clarifications or suggestions. |
Please see the Figure 3, and Table 1 and 2. |
|
- The results section could be clearer, particularly with improved visual aids and more concise presentation of the outcome data. Further controls could include stratifying patients by additional variables eg preop health status or severity of injury. |
A revision of the results section has been conducted, making it clearer, including the improved organization of tables and figures.
The grade of neuropathy and therefore the severity of injury was reported in Table 2. |
Please see the results section with suggested changes made. |
|
- While the conclusions are supported by the findings, emphasizing the study's limitations, especially the small sample size, would enhance the discussion. |
This point has been included among the study’s limitations and provides us with the opportunity to continue data collection in order to update the findings already highlighted in this study. |
“Lastly, the small sample size restricted our ability to perform subgroup analyses based on the specific types of complex elbow dislocation, which might have provided more detailed insights into the effectiveness of different treatment strategies. Additionally, the limited diversity and small size of the patient cohort may have impacted the results' statistical power and generalizability. Future studies should aim to include a more extensive and diverse patient population to improve the robustness and applicability of the findings across broader demographics.“ |
|
- The references are appropriate and cover a range of relevant studies in the field however, authors should consider adding more recent studies or meta-analyses to reinforce the discussion on nerve transposition outcomes |
We have added recent references as suggested. |
Please see the added references:
- Baltassat, A., Baldairon, F., Berthe, S., Bellier, A., Bahlouli, N., & Clavert, P. Creation of a replicable anatomic model of terrible triad of the elbow. J Orthop Surg Res. 2024;19(1):638. - Soderlund, T., Zipperstein, J., Athwal, G. S., & Hoekzema, N. Monteggia Fracture Dislocation. J Orthop Trauma. 2024;38(9S):S26-S30. - Giai Via, R., Faccenda, C., Artiaco, S., Dutto, E., Lavia, A. D., Massè, A., & Battiston, B. Functional and subjective outcomes after surgical management of complex elbow dislocations: a retrospective study. Eur J Orthop Surg Traumatol. Published online September 20, 2024. doi:10.1007/s00590-024-04103-5. - Li, T., Yan, J., Ren, Q., Hu, J., Wang, F., Xiao, C., & Liu, X. Efficacy and safety of anterior transposition of the ulnar nerve for distal humerus fractures: A systematic review and meta-analysis. Front Surg. 2023;9:1005200. |

Reviewer 2 Report
Comments and Suggestions for Authors
The authors present a manuscript in which reference is made to a single-center retrospective study for ulnar nerve management in elbow dislocations. Regarding the manuscript, I suggest the following changes:
1. In the abstract, include a description of the abbreviations used.
2. In the background development, include a representative figure that illustrates the anatomical composition of the elbow, its ligaments and nerves, and the main complications observed.
3. How could ulnar nerve neurolysis be described?
4. Place Figure 1 after mention in the development of the text (the description of Table 1 appears before Figure 1).
5. The flowchart contains overlapping characters in the corresponding descriptions. It is suggested that the font size be increased.
6. Place Table 1 after its mention in the development of the text.
7. Figure 2 should be considered as Table 3 and elaborated with the characteristics of a table.
8. Consider including current references regarding ulnar nerve neuropathy (Hannaford A et al., PMID 38697734).
Comments on the Quality of English LanguageThe quality of the English in the manuscript is acceptable.
Author Response
|
Numbered Reviewer Remark and Manuscript Line Number |
Author Response |
Revised Manuscript Line Number and Text Change |
|
The authors present a manuscript in which reference is made to a single-center retrospective study for ulnar nerve management in elbow dislocations. Regarding the manuscript, I suggest the following changes: 1. In the abstract, include a description of the abbreviations used. |
Suggested corrections have been made in the text. |
Please see the edited text in the abstract:
“Mayo Elbow Performance Score (MEPS), Disabilities of Arm, Shoulder, and Hand (DASH) score, and Oxford Elbow Score (OES).” |
|
2. In the background development, include a representative figure that illustrates the anatomical composition of the elbow, its ligaments and nerves, and the main complications observed. |
We have provided the required figure. |
Please see the Figure 1. |
|
3. How could ulnar nerve neurolysis be described? |
By ‘neurolysis’ we mean ‘external neurolysis’, the procedure that consists of exposing the nerve by freeing it from the sheath and any surrounding scar tissue. |
/ |
|
4. Place Figure 1 after mention in the development of the text (the description of Table 1 appears before Figure 1). |
As suggested, the order of succession of figures and tables has been arranged. |
/ |
|
5. The flowchart contains overlapping characters in the corresponding descriptions. It is suggested that the font size be increased. |
The correction suggests have been made. |
Please see the Figure 2. |
|
6. Place Table 1 after its mention in the development of the text. |
The correction suggests has been made. |
See the manuscript. |
|
7. Figure 2 should be considered as Table 3 and elaborated with the characteristics of a table. |
Figure 2 has been modified to make it suitable as a flowchart, and for this reason, it has been kept as a figure but with a now correct format. Should you prefer it to be converted into a table, we will do so, but it may not be well formatted. |
Please see the Figure 2. |
|
8. Consider including current references regarding ulnar nerve neuropathy (Hannaford A et al., PMID 38697734). |
We have added recent references as suggested. |
Please see the added reference:
Hannaford, A., & Simon, N. G. Ulnar neuropathy. Handb Clin Neurol. 2024;201:103-126. |

Reviewer 3 Report
Comments and Suggestions for Authors
7 pts who had neuropathy after surgery which did not have transposition. Your technique does release the cubital tunnel and proximally. Was the post surgery neuropathy due to instability?
Author Response
|
Numbered Reviewer Remark and Manuscript Line Number |
Author Response |
Revised Manuscript Line Number and Text Change |
|
7 pts who had neuropathy after surgery which did not have transposition. Your technique does release the cubital tunnel and proximally. Was the post surgery neuropathy due to instability? |
Dear Reviewer, thank you for giving us the opportunity to clarify that aspect. None of the post-operative neuropathies were correlated with instability. This is probably due to the fact that in patients in whom ulnar nerve inability was observed during intra-operative flexion-extension maneuvers a nerve anteposition was directly performed.
|
/ |

Round 2
Reviewer 2 Report
Comments and Suggestions for Authors
The authors integrated the suggestions to make the article more understandable and fluent. As a last recommendation, revise the English edition in the section of Figure 2 (in the flowchart, correct the word “enrolment”).

The quality of the English is acceptable; a final manuscript revision is suggested.
Reviewer 3 Report
Comments and Suggestions for Authors
nice revision